# Advances in Two-Dimensional Magnetic Semiconductors via Substitutional Doping of Transition Metal Dichalcogenides

**DOI:** 10.3390/ma16103701

**Published:** 2023-05-12

**Authors:** Mengqi Fang, Eui-Hyeok Yang

**Affiliations:** 1Department of Mechanical Engineering, Stevens Institute of Technology, Hoboken, NJ 07030, USA; mfang3@stevens.edu; 2Center for Quantum Science and Engineering, Stevens Institute of Technology, Hoboken, NJ 07030, USA

**Keywords:** transition metal dichalcogenides, substitutional doping, magnetic property

## Abstract

Transition metal dichalcogenides (TMDs) are two-dimensional (2D) materials with remarkable electrical, optical, and chemical properties. One promising strategy to tailor the properties of TMDs is to create alloys through a dopant-induced modification. Dopants can introduce additional states within the bandgap of TMDs, leading to changes in their optical, electronic, and magnetic properties. This paper overviews chemical vapor deposition (CVD) methods to introduce dopants into TMD monolayers, and discusses the advantages, limitations, and their impacts on the structural, electrical, optical, and magnetic properties of substitutionally doped TMDs. The dopants in TMDs modify the density and type of carriers in the material, thereby influencing the optical properties of the materials. The magnetic moment and circular dichroism in magnetic TMDs are also strongly affected by doping, which enhances the magnetic signal in the material. Finally, we highlight the different doping-induced magnetic properties of TMDs, including superexchange-induced ferromagnetism and valley Zeeman shift. Overall, this review paper provides a comprehensive summary of magnetic TMDs synthesized via CVD, which can guide future research on doped TMDs for various applications, such as spintronics, optoelectronics, and magnetic memory devices.

## 1. Introduction

The van der Waals crystals are two-dimensional (2D) material systems with atomically thin, layered structures. Recent discoveries in 2D materials include ferromagnetism in atomically thin layers of chromium-based alloys [1]. For example, Cr_2_Ge_2_Te_6_ demonstrates a ferromagnetic order that remarkably influences the transition temperature, even with small magnetic fields [2]. Mechanically exfoliated monolayer chromium triiodide (CrI_3_) also achieves ferromagnetic properties with out-of-plane spin orientation under 45 K [3]. While these van der Waals ferromagnets remain either metallic or insulating, spintronics and solid-state quantum information science applications benefit from semiconductors with ferromagnetic properties known as dilute magnetic semiconductors (DMSs). While these DMS materials have been extensively researched for decades in their bulk form, of which typical representatives are (Cd, Mn)Te, (Zn, Co)O, and (Pb, Eu)S [4], transitioning bulk dilute magnetic semiconductors (DMSs) into the 2D field remains challenging [5,6]. For example, MnTe has shown antiferromagnetic behavior that can be transverse to paramagnetic by reducing the thickness of the material to the few-layer regime via exfoliation [7]. Despite its promising properties in its monolayer form, achieving a monolayer has proven to be challenging due to the strong interlayer bonding between the layers [8].

Several theoretical studies have predicted that DMSs based on TMDs would exhibit ferromagnetic behaviors even at room temperature, which is a fundamental requirement for practical applications [9,10,11,12,13,14]. For example, delaminating a non-van der Waals (vdW) magnet, MnSe_2_ has been reported to have long-range magnetic ordering due to the ferromagnetically coupled Mn atoms, where the Curie temperature reaches room temperature with a 5% applied strain [15,16]. First-principles calculations anticipate that Fe- and V-doped MoSe_2_ would exhibit room-temperature out-of-plane ferromagnets at high atomic substitutions [9].

Several experimental studies have utilized ex-situ techniques to dope transition metal atoms (V, Cr, Fe, etc.) in post-processing techniques into TMDs [17]. Adsorption and intercalation of transition metals and non-metal elements have introduced para- or ferromagnetism into MoS_2_ crystals for spintronic applications [18,19]. While these approaches have shown promising routes in creating novel magnetic properties in 2D materials, they resulted in a Curie temperature well below room temperature or random local clustering of magnetic precipitations. Other attempts to incorporate magnetic atoms directly in situ during growth relied upon converting bulk TMD crystals into DMSs. For example, Fe was doped in SnS_2_ bulk crystal via a direct vapor phase method, while mechanical exfoliation was required to reveal the van der Waals DMS [20]. Recently, the direct in-situ growth of transition metal-doped TMDs via chemical vapor deposition (CVD) overcame some of the difficulties of post-processing conversion approaches [21]. For example, rhenium (Re), vanadium (V), or iron (Fe) were proven to substitute either molybdenum (Mo) or tungsten (W) sites in monolayer MoS_2_, WS_2_, and WSe_2_ crystalline [22]. 

Two-dimensional DMS offers intriguing applications combining magnetism with semiconducting properties, including magneto-optoelectronic devices for spintronics, optics, and the quantum field. An Fe-doped SnS_2_ homojunction device was reported to have an unsaturated magnetoresistance of 1800% [23]. A giant magnetic circular dichroism was demonstrated by the magneto-optical effects in monolayer, bilayer, and trilayer DMSs, leading to a significant MO Kerr rotation and Faraday rotation angles. [24]. Reports on device applications using 2D magnets include the magnetic tunnel junction in heterostructure devices with 2D layers of Fe_3_GeTe_2_ [25], Cr_2_Ge_2_Te_6_ [26], and CrI_3_ [27]. Ferromagnetic properties in doped TMDs provide potential opportunities for many groundbreaking applications. Here we provide our perspective on the synthesis, characterization, and magnetic properties of magnetic 2D transition metal dichalcogenides (TMDs) formed via substitutional doping with CVD.

## 2. Effects of Magnetic Doping in 2D TMDs

Ferromagnetism in 2D limits can be induced by substituting the atoms in the d-block of the periodic table with the original transition metal atoms or chalcogen atoms, creating 2D DMS materials. Transition metal dopants and transition atoms in TMD lattices are polarized primarily by their localized 3d and 5d electrons. For instance, based on the density functional theory, a hybridization of localized transition metal-3d, delocalized Se-4p, or W-5d states results in electronic structures of spin-up and spin-down channels in the WSe_2_ lattice, leading to the ferromagnetic coupling between the doped transition metal atom spins and Se or W spins [11]. Similarly, due to low formation energy, Mn dopants in MoS_2_ tend to replace Mo atoms in Mn-doped MoS_2_ crystals [28]. Upon substitution, a portion of the Mn dopants’ valence electrons (3d^6^4s^1^) move to the nearest S atoms, and the remaining valence electrons cause spin polarization, creating a magnetic moment [29]. Dopant concentrations also affect the magnetic-exchange coupling related to ferromagnetic and antiferromagnetic properties. Although detecting the concentration of dopants in all areas of TMDs remains challenging, first principal calculations have shown that the clustering and aggregation of dopants can significantly impact the resulting magnetic moment [9,30]. Prior research has demonstrated that isolated Mn doping at the Mo site of MoS_2_ defects often cluster together [12,31], the clusters of Mn-Mo defects with low formation energies can preserve magnetic ordering, while Mn-Mn clusters lead to strong local lattice distortion [32]. When two Mn dopants are close to each other, the ferromagnetic state is preferred. However, the Mn dopants exhibit strong spin polarization and antiferromagnetic coupling with the S atoms when their separation distance is less than 16.53 Å, resulting in the same spin-polarization direction [29,33]. MoSe_2_ can also be doped with two dopant atoms (Ti, V, Mn, Cr, Fe, Co, Ni), resulting in a weak magnetic/non-magnetic state, as shown in Figure 1a. Figure 1b–f illustrate the magnetically ordered state of different dopant atoms. V, Cr, Mn, Fe, and Co have magnetic moments exceeding 1.6 μB per dopant [9]. The electronic band structures also significantly change in d-orbital hybridization after Fe is substitutionally doped in monolayer MoS_2_ with a further enhanced valley Zeeman splitting (Figure 1g,h) [34]. The effect of hybridization between the conduction band and d band in the transition metal dopants and the TMD lattice leads to ferromagnetism and antiferromagnetism in 2D doping TMD monolayers. In monolayer MoS_2_, Mn dopants have been found to induce ferromagnetic coupling through a double-exchange mechanism, which is mediated by the nearest neighboring sulfur atoms. Conversely, in bilayer MoS_2_, the magnetic exchange coupling between Mn dopants is influenced by the interplay of multiple interactions, including double-exchange, direct-exchange, and superexchange interactions [29]. The contribution of these interactions to the magnetic behavior in multilayer TMDs is not fully understood. 

Magnetic dopants can significantly impact the stability of TMDs in different phases (2H, 1T, 1T’). The most stable crystal phase is determined by the d-orbital electron numbers of the transition metal [35]. Most studies have investigated the impact of random doping in the 2H phase of trigonal prismatic TMDs [36]. Re-doped MoS_2_ nanosheets show a transformation from the 2H to 1T phase in both the Re and Mo coordinating structures, where the 1T ratios of both Mo and Re atoms were found to increase with the increasing Re-doping concentration, inducing the ferromagnetic ordering and stabilizing the metastable crystal phase [37]. In the 1T’ phase, which is a metastable phase of TMDs, magnetic doping can alter the energy landscape of the material [38]. 

In 3D DMS materials, the magnetic moments of impurities are typically coupled through the Ruderman–Kittel–Kasuya–Yosida (RKKY) interaction, which is mediated by the conduction electrons in the semiconductor host [39,40,41]. As a result, ferromagnetism is typically observed only when the Mn concentration is above a critical value, called the percolation threshold [41,42]. They form a connected network of magnetic moments that can collectively exhibit ferromagnetism with a high concentration of Mn atoms [43,44]. In Mn-doped GaAs and Mn-doped InAs, the magnetic ions are randomly distributed throughout the lattice, leading to a percolative ferromagnetic phase transition as the temperature is lowered [45,46]. In contrast, some magnetically doped 2D TMDs exhibit magnetic ordering up to room temperature [9,47]. While in 2D DMS materials, the nature of the magnetic interactions can be more complex due to the reduced dimensionality of the system. This confinement leads to a more significant interaction between the dopant atoms, which enhances the formation of ferromagnetic clusters even at low doping levels [48,49].

## 3. Synthesis via Chemical Vapor Deposition (CVD)

This perspective article focuses on the CVD-grown 2D DMS. While several synthesis strategies have been demonstrated, including physical vapor deposition (PVD), atomic layer deposition (ALD), molecular beam epitaxy (MBE), and pulsed laser deposition (PLD) [50], CVD growth is commonly used to synthesize 2D TMDs [51]. Typically, the transition metal source sits in the middle of the heating zone, and the chalcogen source is placed upstream of the carrier gas (typically Ar or N_2_, usually combined with H_2_). Heating the furnace to 700–1000 °C causes the vaporization of transition metal and chalcogen sources, leading to the formation of nanoparticles diffusing in the carrying gas, which then condenses on the substrate. Doping can be achieved by diffusing dopants into TMDs during precursor vaporization, making the CVD growth an effective means to generate TMD alloys [52,53].

### 3.1. LPCVD Growth and Doping

#### 3.1.1. Solid Source-Based Growth and Doping of TMDs

The solid source-based TMD growth requires a mixture of solid dopant powder, such as elemental substance, metal salt, and metal oxide, with a transition metal source. This growth is realized based on the simple mixing of precursors, which does not involve additional multiplex pre-operating procedures or complex pressure and flow control. For example, Fe-doped MoS_2_ can be synthesized using the MoO_3_ and FeS_2_ powders with different Fe-to-Mo molar ratios of 0.12, 0.24, 0.48, and 0.4 (Figure 2j,k) [34]. Cr-doped WTe_2_ crystals were grown via a two-step Te flux. The W and Cr powders and Te granules were mixed and heated at 1050 °C for 24 h, then Cr atoms were diffused into layered semimetal Td-Wte_2_. Figure 2f,g illustrate the corresponding elemental mapping images. [54]. Mn atoms were successfully in situ-doped into the MoS_2_ lattice by introducing the Mn_2_(CO)_10_ precursor upstream, as shown in Figure 2a–c, whereby the incorporation of Mn in MoS_2_ was enabled by inert substrates, but not the reactive surface terminations, which led to the formation of defective MoS_2_ (Figure 2d,e) [21]. Another study used Fe_3_O_4_ particles cast on the substrate as the dopant source, permitting substitutional doping of Fe atoms into Mo or W sites in TMD crystals [47,55]. Figure 2h–k show the schematics and STEM images of in situ-doped monolayer Fe:MoS_2_. Intrinsic defects in doped TMD monolayers can also induce magnetic moments, and the chemical potential of the chalcogen influences the formation energy of these defects. This chemical potential can be modulated by adjusting the ratio of chalcogen to transition metal during CVD growth [56].

#### 3.1.2. Liquid-Assisted Growth and Doping of TMDs

The liquid-assisted growth uses a metal salt aqueous solution or metal suspension directly spun on the substrate. Although it requires more steps than using the solid source, it enables a more homogeneous mixing of precursors, contributing to more uniform doping in the growing TMD lattice than the solid source-based doping. For example, the V-doped monolayer WSe_2_ demonstrated ferromagnetic domains by controlling the atomic ratio of V to W in a precursor solution from 0.1% to 40%, as illustrated in Figure 3a,b. Furthermore, in this growth, the doping concentration was controlled from 1% to a relatively high atom weight percent [57]. Lastly, the one-pot mixed-salt-intermediated CVD method was employed to synthesize single-crystal magnetic group VIII transition metal-doped MoSe_2_, which demonstrated excellent controllability and reproducibility (Figure 3c), and Fe-doped MoSe_2_ monolayers were obtained by adjusting the FeCl_3_/Na_2_MoO_4_ ratios to control Fe-doping concentrations ranging from 0.93% to 6.10% [58]. 

### 3.2. MOCVD Growth and Doping

The metal–organic chemical vapor deposition (MOCVD) uses a pulsed precursor gas vapor source. It is a highly complex process for growing wafer-scale 2D crystalline layers with an excellent uniformity of deposition rates, dopant concentrations, and layer thickness [59], which are crucial for practical nanoelectronics and optoelectronics applications [60]. For example, large-area Nb-doped monolayer MoS_2_ was grown by employing molybdenum hexacarbonyl (Mo(CO)_6_) and diethyl sulfide (DES) as Mo and S precursors. Figure 3d,e show the schematic of the MOCVD process and as-grown Nb-MoS_2_ [60]. Multilayer V-doped WSe_2_ has been demonstrated on quasi-freestanding epitaxial graphene on 6H-SiC (Figure 3j–l) [61]. WSe_2_ films with coalesced 2D surfaces were doped with 0.5–1.1% of Re atoms in situ in another systematic study: a precisely controlled doping concentration was achieved by tuning the precursor partial pressure, with decreased domain size and increased doping concentration. The Re dopant and decreased domain size are illustrated in Figure 3f–i [62].

**Figure 3 materials-16-03701-f003:**
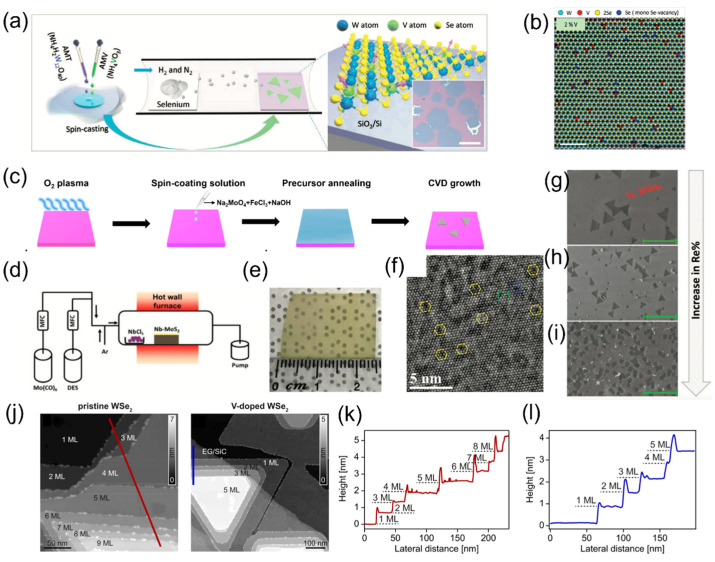
(**a**) Schematic of the V-doped WSe_2_ synthesis using a liquid precursor and mixing W with V. Scare bar: 50 µm. (**b**) False-color Wiener-filtered STEM images of 2% V-doped WSe_2_. Scare bar: 5 Å [57]. (**c**) Process sequence of the Fe-doped MoSe_2_ using a one-pot mixed-salt-intermediated CVD method [58]. (**d**) Schematic of the MOCVD setup. (**e**) Optical image of uniformly grown Nb-doped MoS_2_, the scale bar is 1 cm [60]. (**f**) HRTEM image shows a periodic atomic arrangement where Re atoms are introduced in the WSe_2_ lattice, indicated by yellow circles. (**g**) SEM images of Re:WSe_2_ monolayers with 0.5% (**h**) and 1.1% (**i**) doping concentrations. Re-doped WSe_2_ films reveal significant changes in the surface morphology following the introduction of Re atoms [62]. (**j**) Scanning tunneling microscopy (STM) image (VB = 1.6 V) of undoped and V-doped multilayer WSe_2_, with the number of WSe_2_ layers labeled, providing valuable insights into the electronic and structural properties of WSe_2_ materials. (**k**,**l**) Height profiles along the lines shown in the corresponding STM images of undoped (**k**) and V-doped WSe_2_ (**l**) [61].

As summarized in Table 1, solid source-based thermal CVD growth and doping are convenient means to grow various TMDs by vaporizing the metal-oxide precursors [52]. The liquid phase-assisted CVD growth and doping can achieve a more homogeneous mixing of dopants and transition metal sources at the molecular level than the solid source-based CVD growth [58]. However, the asynchronous evaporation of these dopants and precursors during CVD can lead to heterogeneous element distribution and aggregation, impeding material characterization and limiting the performance of resulting devices. MOCVD enables better control of the precursor supply in growing TMDs than thermal CVD. The metal–organic precursors can easily be turned to the vapor phase [63], permitting a lower growth temperature than thermal CVD growth. However, residual carbon contamination may occur with some precursors in the grown 2D film [64]. 

**Table 1 materials-16-03701-t001:** Comparison of the growth methods.

Material Type	Dopant	Synthesis Methods	Size	Thickness	Doping Concentrations	Semiconductor Type after Doping	Ref
MoS_2_	Co, Cr	Solid source CVD	~20 um	monolayer	Co 1% Cr 0.3%	p-type	[65]
MoS_2_	Fe	Solid source CVD	~20 um	monolayer	0.3~0.5%	-	[47]
MoS_2_	Fe	Solid source CVD	~30 um	monolayer	0.40%	-	[34]
WSe_2_	V	Liquid-phase assistant CVD	>50 um	monolayer	0.5–10%	p-type	[57]
Td-Wte_2_	Cr	Two-step Te flux CVD	~1 um	bulk	2%	-	[54]
MoS_2_	Mn	Solid source CVD	~200 nm	monolayer	2%	-	[21]
WS_2_	V	Liquid-phase assistant CVD	-	monolayer	0.4–12%	p-type	[49]
MoSe_2_	Fe	Liquid-phase assistant CVD	~40 um	monolayer	0.93–6.1%	n-type	[58]
MoTe_2_	Cr	Solid source CVD	>1 um	2H bulk	2.1–4.3%	p-type	[66]
MoS_2_	Re	Solid source CVD	~15 um	monolayer	1%	n-type	[22]
MoS_2_	Nb	MOCVD	Wafer-scale	monolayer	5%	-	[60]
Wse_2_	V	MOCVD	-	monolayer to multilayer	0.44%	p-type	[61]
Wse_2_	Re	MOCVD	450–500 nm	monolayer	0.5–1.1%	n-type	[62]

## 4. Optical, Magnetic, and Other Properties 

### 4.1. Optical Properties

The electron–phonon coupling strengths are impacted by the modification of the A’_1_ vibrational mode (out-of-plane vibration of chalcogen atoms), which is caused by the electron density shift resulting from the substitutional doping [67,68,69]. For example, in Figure 4a, Fe-doped MoS_2_ showed the n-type doping effect, which is revealed by a blue shift of the A_1g_ peak compared to undoped MoS_2_ [34]. Co- or Cr-doped MoS_2_ showed a moderate blue shift of the A_1g_ peak, while unaffecting the E^1^_2g_ peak (in-plane vibration mode of chalcogen atoms), showing the sensitivity of A_1g_ peak to dopant atoms [65]. The density and type of carriers can be modified after doping, affecting the PL spectra of the materials dramatically. For example, the Re doping of Wse_2_ showed a PL redshift by 40 meV and quenching by a factor of two, caused by non-radiative trion recombination [62]. A similar redshift was observed in Nb-doped MoS_2_ with a ~150% intensity enhancement due to impurity state formation caused by the Nb atoms [60]. Moreover, Fe-doped WS_2_ showed a PL blueshift by 13 ± 0.5 meV and PL quenching by 40%. Fe-doped MoS_2_ exhibited a PL quenching of 35% and a redshift of 29 ± 0.5 meV [55]. The difference between Fe:MoS_2_ and Fe:WS_2_ is attributed to the interaction of neutral excitons and negative/positive trions [70]. The detrimental effect of dopant-induced defects on the optical properties of 2D materials can be mitigated through the growth of dual dopant monolayers, thereby tuning these properties. For example, incorporating Co and Cr atoms in MoS_2_ monolayers suppresses nonradiative recombination, significantly improving the photoluminescence intensity [65].

### 4.2. Magnetic Properties

In TMD crystals, direct interband transitions at the K and K′ valleys are coupled exclusively with the left and right circularly polarized light [71]. Furthermore, the inversion symmetry enhanced by dopant atoms leads to the existence of a magnetic moment within the orbit of the electrons. Therefore, the valley magnetic moment can be controlled in monolayer TMDs by circularly polarized light and the out-of-plane magnetic field, permitting control of the valley degree of freedom [72]. Strong circular dichroism (CD) was observed in the Fe-related peak at 2.28eV from the Fe-doped monolayer MoS_2_ when opposite circularly polarized lights were applied. The magnetic circular dichroism (MCD), as a function of increasing and decreasing magnetic fields, also showed a distinct hysteresis loop in Figure 4i,j [47]. Figure 4d–f show the valley Zeeman shift of Fe-doped MoS_2_. At zero field, the PL spectra were completely overlapped when σ− and σ+ light were applied; however, they split at high magnetic fields with inverse shifting directions for opposite magnetic field directions [34]. 

The Goodenough–Kanamori–Anderson rules state that superexchange can lead to ferromagnetic short-range coupling between overlapping orbitals [73]. In Nb and Co co-doped WSe_2_, the magnetic signal was strongly enhanced with the doping of Co at 4%. The saturation magnetization (Ms) reached 63.6 emu cm^−3^ at 10 K, as shown in Figure 4c [48]. The magnetization versus magnetic field strength (M−H) test for Cr-doped MoTe_2_ showed a significant enhancement in an out-of-plane magnetic field with a Curie temperature around 275 K. The maximum Ms value occurred at 4.78 emu g^−1^ for the bulk 4.3% Cr-doped MoTe_2_ [66]. Semiconducting V-doped WSe_2_ monolayers showed temperature-dependent ferromagnetic domains with long-range order, as shown in Figure 4g,h. Magnetic force microscopy (MFM) was used to detect the ferromagnetic domain stripes with a solid contrast, where the splitting and merging phenomenon of domains was observed with the temperature changes from low to high temperature. The magnetic domain walls retained the vestige even at 420 K. Meanwhile, V-doped WSe_2_ showed p-type semiconductor behavior after doping [57]. The reported magnetization and coercivity in doped TMDs corresponding to measurement temperatures are summarized in Table 2.

**Table 2 materials-16-03701-t002:** The magnetization and coercivity in different transition metal-doped TMDs.

Material	Dopant Concentration	Saturation Magnetization	Coercivity	Temperature	Reference
Fe-doped SnS_2_	2.10%	3.49 × 10^−3^ emug^−1^	400 Oe	300K	[20]
Co-Cr-doped MoS_2_	1% Co, 0.3% Cr	0.4 emu cm^−3^	100 Oe	300K	[65]
V-doped MoTe_2_	0.30%	0.6 μemu cm^−2^	-	300K	[17]
Co- and Nb-doped WSe_2_	4%	60.62 emu g^−1^	1.2 k Oe	10K	[48]
Cr-doped Td-WTe_2_	1%	4.20 emu g^–1^	-	3K	[54]
V-doped WS_2_	2%	2.85 × 10^−5^ emu cm^−2^	180 Oe	50K	[49]
Mn-doped MoSe_2_	6.10%	2 × 10^−5^ emu g^−1^	-	300K	[58]
Cr-doped 2H-MoTe_2_	2.50%	4.78 emu g^−1^	6322 Oe	3K	[66]
V-doped MoS_2_	5%	0.067 emu g^−1^	1870 Oe	10K	[74]

### 4.3. Other Properties

Recent research suggests that doping can modify the interlayer chemical bond in layer-structured materials, allowing access to layer-dependent transport properties [75]. In the case of MoS_2_, vanadium (V) doping has been shown to enhance in-plane conductivity and improve carrier concentration [76]. Additionally, dopants can activate out-of-plane interactions between adjacent layers, further tailoring the electrical carrier transport behavior of bilayers [77]. Doping of Mn in MoS_2_ also results in improved electrical contact in field effect transistors due to the Fermi level shift towards the conduction band [78]. However, the mobility slightly decreases when doping Cr in monolayer MoS_2_, which is attributed to the change in intrinsic defect density [22,57]. When reacting with Ni atoms, the intrinsic defects and distorted lattice improve the carrier injection, modifying the electronic properties [79], as summarized in the previous review [80].

In addition, TMDs show excellent catalytic properties owing to their high surface area and edge sites, which are highly active for catalytic reactions. These properties make TMDs promising candidates for catalytic applications in energy conversion, environmental remediation, and chemical synthesis [81]. For example, hydrogen evolution reaction (HER) using TMDs has been studied due to their remarkable catalyst activation performance compared to bulk forms [82]. Using elemental doping of TMDs has emerged as an effective approach to tune the electronic structure of catalytic sites, optimizing the Gibbs free energy of hydrogen adsorption (ΔGH) to promote HER kinetics [82,83]. Notably, various dopants have been found to activate the inactive basal plane of 2D TMDs into active ones by creating defects [84]. Previous studies have already summarized the discussion on the catalytic properties of dopant TMDs, indicating their potential for various applications [85]. For a more comprehensive review of the application of TMDs in different structures, such as heterostructures or nanoribbons, we refer the readers to [86,87,88]. 

## 5. Conclusions

Doping of TMDs is a popular topic since the external atoms can bring or enhance multiple properties of origin TMDs. In magnetic doping, the promoted magnetic properties due to the large atomic mass of doped transition metals can shed new light on their potential applications. This review has introduced different doping methods via CVD, including solid source-based doping, liquid-assisted doping, and MOCVD methods. Dopant-induced optical and magnetic properties of doped TMDs were also summarized. 

The exploration of magnetically doped TMDs poses several challenges. First, the uniform and large-area TMD synthesis remains challenging due to the difficulty in controlling the flow of vaporized precursors and uncontrollable domain size [36]. Though room temperature ferromagnets were demonstrated, most of the magnetic TMDs show low Curie temperatures hampering the practical applications of these materials. Optimizing the exchange interaction and magnetic anisotropy would be key parameters to enhance the critical temperatures [1]. Another challenge includes randomly distributed dopants in the crystal lattice, causing localized magnetic fields. The long-range behavior of magnets is attributed to the finite lattice size, concentrated and ordered positions of magnetic ions, and different saturation magnetizations caused by different kinds of dopants. [9]. The co-doping in TMDs has been demonstrated as a promising approach for tuning the properties of 2D materials. Incorporating two dopant elements in the material can suppress nonradiative recombination, enhancing optical properties, and inducing ferromagnetic ordering. In addition, the electronic interactions between the dopants can further modify the band structure and energy levels of the materials, providing additional opportunities for tailoring their electrical and magnetic properties. The topological boundary states, magnetic proximity effects, and van der Waals heterostructures are emerging fields utilizing magnetic 2D TMDs. With groundbreaking research in the coming years, transition metal-doped TMDs are promising materials in optomagnetism, spintronic devices, and quantum information science.

## Figures and Tables

**Figure 1 materials-16-03701-f001:**
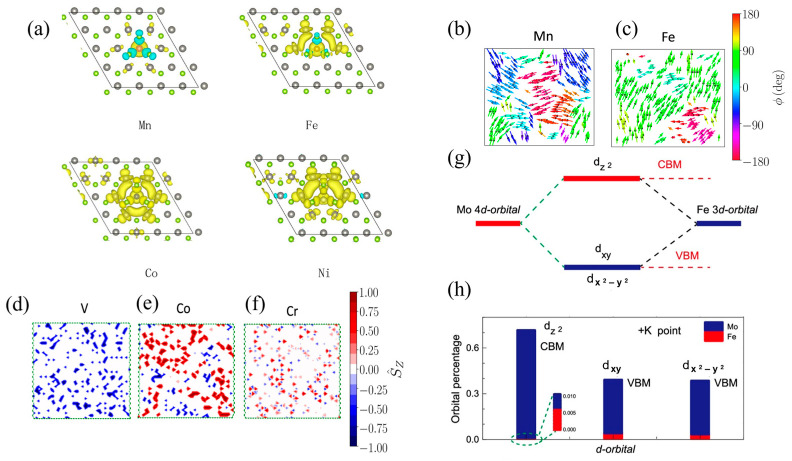
(**a**) Spin density for a single transition metal dopant atom (Mn, Fe, Co, Ni) in a monolayer WSe_2_ supercell with a 6.25% doping concentration of transition metals. The gray and green balls represent W and Se atoms, respectively. Yellow and cyan isosurfaces represent positive and negative spin densities [9]. (**b**–**f**) The magnetically ordered state of the Mn- (**b**), Fe- (**c**), V- (**d**), Co- (**e**), and Cr-doped (**f**) samples of MoSe_2_ with an atomic substitution of 15%, at a temperature of 5 K [9]. (**g**) Diagram of Fe and Mo atoms hybridized in d-orbitals for a 5 × 5 MoS_2_ supercell with one Fe atom instead of one Mo atom. (**h**) Percentage of hybridized d-orbitals of the Fe-doped monolayer MoS_2_ for each isolator d-orbital. The valence band maximum is predominantly composed of the d_x_^2^_−y_^2^ and d_xy_ orbitals of both Mo and Fe, whereas the conduction band minimum is primarily dominated by the d_z2_ orbitals. Although the Fe d-orbitals contribute to the orbital hybridization, their relative fractions are comparatively small compared to Mo d-orbitals [34].

**Figure 2 materials-16-03701-f002:**
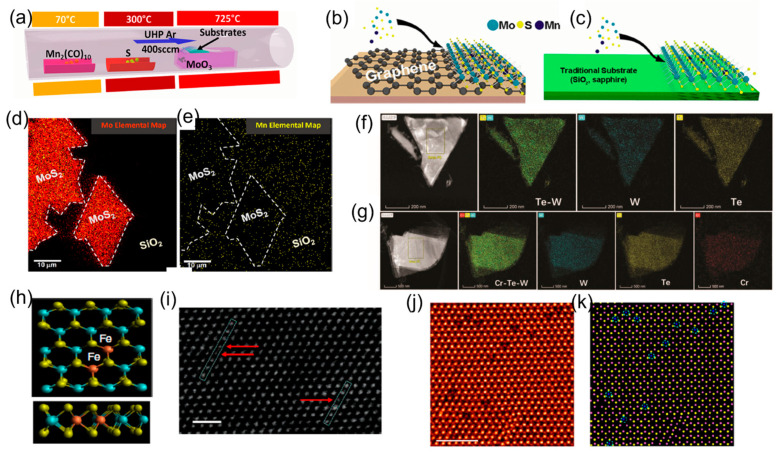
(**a**) Schematic of powder vaporization techniques used to synthesize manganese-doped and undoped MoS_2_ [21]. Schematic of the growth substrates, including (**b**) graphene and (**c**) insulating substrate (sapphire and SiO_2_). (**d**) Element analysis of Mn-doped MoS_2_ with (**e**) Mn concentrations (yellow pixels) that are equal (or higher) to those found in the MoS_2_ areas, which suggests that Mn may be bound to the substrate surface rather than incorporated into it [21]. (**f**,**g**) STEM-HAADF images of (**f**) pure WTe_2_ and (**g**) Cr_0.02_–WTe_2_ samples with their corresponding element mapping images [54]. (**h**) Schematics of the monolayer Fe:MoS_2_ with S, Mo, and Fe atoms denoted by green, blue, and red spheres. (**i**) Contrast-corrected STEM image of monolayer Fe:MoS_2_. The arrow shows iron atoms position in crystal lattice. Scale bar: 1 nm. [47]. (**j**) Typical atomic-resolution ADF-STEM image of Fe-doped monolayer 1H-MoS_2_. (**k**) Corresponding atomic mode showing the distribution of Fe dopant atoms in (**j**). Blue, pink, and yellow spheres represent Fe, Mo, and S atoms. Scale bar: 2 nm [34].

**Figure 4 materials-16-03701-f004:**
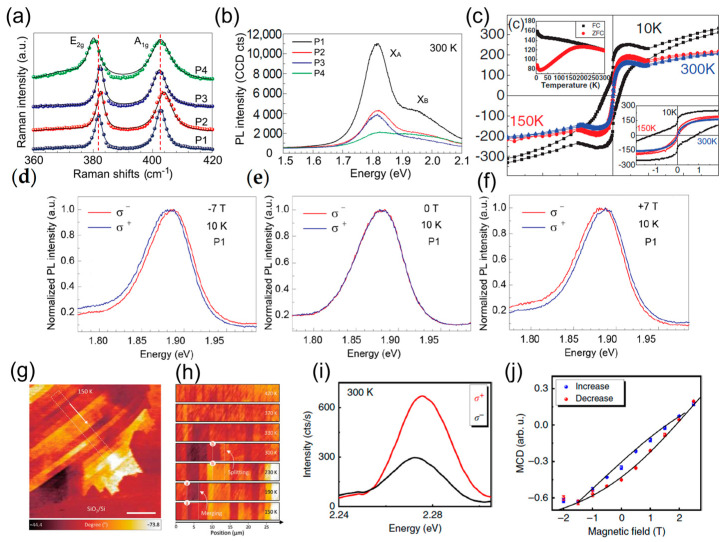
(**a**) Raman and (**b**) PL spectra of Fe-doped monolayer MoS_2_ at room temperature [34]. (**c**) The hysteresis loop of Nb at 1% and Co at 4% co-doped WSe_2_, along with zero-field-cooled (ZFC) and field-cooled (FC) curves shown in the top left inset, and the magnified M–H curves in the bottom right inset, provide a characterization of the magnetic properties at different temperatures [48]. (**d**) The normalized raw polarization-resolved valley exciton photoluminescence (PL) spectra of Fe-doped monolayer MoS_2_ without a magnetic field (**e**) and under ±7 T (**d**–**f**) at 10 K [34]. (**g**) MFM phase image of 0.1% V-doped WSe_2_ taken at 150 K. (**h**) Temperature-dependent transition of magnetic domains [57]. (**i**) The red and black spectra represent the Fe-related spontaneous emissions under excitation with opposite circularly polarized light states at 300 K. (**j**) Corresponding magnetic circular dichroism (MCD), as a function of increasing (blue circles) and decreasing (red circles) magnetic fields, recorded at 4 K [47].

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
