# Peer review of "Advances in Two-Dimensional Magnetic Semiconductors via Substitutional Doping of Transition Metal Dichalcogenides"

_materials, 2023, doi:10.3390/ma16103701_

Round 1

Reviewer 1 Report

Authors reviewed the development of 2D TMD magnetic semiconductors. By comparing different CVD strategies and characterization ways, they further proposed some outlook for doping induced room temperature magnetic properties. The reviewed literature are properly clarified. In principle, I recommend this manuscript for publication in Nanomaterials but there are some minor revisons that need to be pointed out.

1. The current topic is rather narrow if only focusing CVD-derived materials. More specific, it should cover other preparation routes like conventional chemical methods. Still, some magnetic semiconductors without doping can also possess long-range magnetic ordering (Mater. Horiz., 2021, 8, 1286). The addition of these discussion part can better appeal to a wide readership.

2. For the first and the last paragraph, they come from the template WORD file. Please delete them.

3. There are very few cases about dual dopant in this review.

4. In table 2, what is the exact meaning about the column of temperature? Curie temperature or measuring tempearture?

5. The citation format of reference 54 is uncorrect.

Reviewer 2 Report

Authors have presented the review on 2D Magnetic Semiconductors via Substitutional Doping of 2 Transition Metal Dichalcogenides. It is interesting and brief review on the topic. Before its acceptance in the journal some points to be addressed:

1. English language should be corrected in the manuscript.

2.Add more recent references in the introduction part and add the section of current status of the study in the introduction part.

3. Add more physical properties of the 2D Magnetic Semiconductors in the characterization part.

4. Add proper section for its other structural form significant for possible applications in the devices.

5. Add transport properties section for the 2D Magnetic Semiconductors.

Reviewer 3 Report

M. Fang and E.-H. Yang have reported a review paper on the doping-introduced magnetic properties of CVD grown 2D TMD materials. The topic of dopants in 2D materials is definitely of interest for the magnetic community, however the presented result of the manuscript is not sufficient for a review paper. In my point of view, a review paper is more than a very short description of different papers in 1-2 sentences. It should contain an in-depth critical review of the literature, where the reader can understand the main phenomena and research directions. The present manuscript is simply too short to achieve the above-mentioned goals. For example, the Liquid assisted growth section (3.1.2) is only 12 lines, while the MOCVD (3.2) is only 11 lines, which include both the description of the methods and the literature data. Besides the length of the manuscript, there are also several important issues related to the magnetic properties, not discussed at all. (For example, the effect of the carrier density on the magnetism). In my opinion the present manuscript needs further extension before possible publication in Materials. My detailed comments and questions are listed as follows:

1, In the whole manuscript the Authors do not refer to the Figures in the main text. This makes even more difficult to understand the briefly described phenomena in the text.

2, In the section 2. (Ferromagnetism in 2D limits) should contain a comparison of the magnetic properties of the 3D DMS materials and the 2D DMS materials. In addition, a general overview of the 2D magnetism beyond the DFT level (cited in the papers) would be also necessary.

3, As I mentioned previously, carrier doping effects are important for several applications. The Authors should discuss the effect of the carrier doping on the magnetism in 2D DMS materials.

4, The main focus of the paper is the comparison of the different CVD growth methods (Table 1.) However, this comparison is only 8 lines, without several important details of grown 2D DMS materials. For example, how the magnetic dopants affect the stability of the 2D materials (2H, 1T, 1T’ phases..)? Do the intrinsic defects in these materials play a major role of the positions and the geometry of the doping atoms? Can the dopants aggregate during the synthesis? How the multilayer structure modifies the magnetic properties? and so forth…

5, Minor comments: Ref [52] and [54] are not correctly cites, and the text between lines 29-37 and lines 286-287 seem to be the remain of the template texts of the journal.

Round 2

Reviewer 3 Report

The authors have satisfactorily answered to my comments and I found that the manuscript has been improved significantly. Consequently, I recommend this work for publication.